# *Wolbachia* infections in natural *Anopheles* populations affect egg laying and negatively correlate with *Plasmodium* development

W. Robert Shaw[1],[*], Perrine Marcenac[1],[*], Lauren M. Childs[2],[3], Caroline O. Buckee[2],[3], Francesco Baldini[4], Simon P. Sawadogo[5], Roch K. Dabiré[5], Abdoulaye Diabaté[5] & Flaminia Catteruccia[1]

The maternally inherited alpha-proteobacterium *Wolbachia* has been proposed as a tool to block transmission of devastating mosquito-borne infectious diseases like dengue and malaria. Here we study the reproductive manipulations induced by a recently identified *Wolbachia* strain that stably infects natural mosquito populations of a major malaria vector, *Anopheles coluzzii*, in Burkina Faso. We determine that these infections significantly accelerate egg laying but do not induce cytoplasmic incompatibility or sex-ratio distortion, two parasitic reproductive phenotypes that facilitate the spread of other *Wolbachia* strains within insect hosts. Analysis of 221 blood-fed *A. coluzzii* females collected from houses shows a negative correlation between the presence of *Plasmodium* parasites and *Wolbachia* infection. A mathematical model incorporating these results predicts that infection with these endosymbionts may reduce malaria prevalence in human populations. These data suggest that *Wolbachia* may be an important player in malaria transmission dynamics in Sub-Saharan Africa.

[1] Department of Immunology and Infectious Diseases, Harvard T.H. Chan School of Public Health, Boston, Massachusetts 02115, USA. [2] Center for Communicable Disease Dynamics, Harvard T.H. Chan School of Public Health, Boston, Massachusetts 02115, USA. [3] Department of Epidemiology, Harvard T.H. Chan School of Public Health, Boston, Massachusetts 02115, USA. [4] Institute of Biodiversity, Animal Health and Comparative Medicine, University of Glasgow, Glasgow G12 8QQ, UK. [5] Institut de Recherche en Sciences de la Santé/Centre Muraz, O1 BP 390 Bobo-Dioulasso 01, Burkina Faso. [*] These authors contributed equally to this work. Correspondence and requests for materials should be addressed to F.C. (email: fcatter@hsph.harvard.edu).

Every year nearly 200 million people contract malaria and around 450,000 people die from the disease, mostly young children in Sub-Saharan Africa[1]. Malaria transmission depends on the complex ecological determinants that drive population dynamics of *Anopheles* mosquitoes, the vectors of human malaria parasites. Measures aimed at the mosquito vector have been the mainstay of malaria control strategies[1,2] and have achieved significant decreases in the global burden of this disease over the last decade, primarily attributable to the widespread use of long-lasting insecticide-treated nets (LLINs)[3]. However, insecticide-based prevention tools are severely threatened by the rapid spread and global distribution of insecticide resistance in anopheline populations[4,5], making the development of new insecticide-free tools for reducing malaria transmission a crucial priority[6].

The use of *Wolbachia* bacteria—intracellular endosymbionts of arthropods and nematodes—has long been suggested for the control of mosquito populations transmitting viral or parasitic pathogens such as dengue fever and malaria, given their profound effects on both insect physiology and pathogen development[7]. These bacteria are best known as active modulators of host reproduction via the induction of mechanisms that promote their rapid invasion of insect host populations. The most commonly observed of these phenotypes is cytoplasmic incompatibility (CI), which prevents uninfected females from producing viable progeny after mating with infected males due to chromosomal segregation defects in the first cellular divisions of the fertilized egg (reviewed in ref. 8). However, infected females are able to produce viable progeny regardless of whether they mate with infected or uninfected males through unknown rescue mechanisms, an effect that favours *Wolbachia*'s spread[9,10]. Additional phenotypes include vertical transmission from mother to progeny, and a female bias in the sex ratio through selective male killing, male feminization and female partheno-genesis (where unfertilized eggs develop as females)[11-13]. Combined with these properties, *Wolbachia* infections in many mosquito species can prevent pathogen development and block disease transmission from vector to human. The list of vector-borne pathogens affected by *Wolbachia* is broad and includes viral pathogens causing dengue, yellow fever, West Nile and Chikungunya, as well as *Plasmodium* parasites[14-18], suggesting a general mechanism of pathogen blocking by an upregulated immune response in the mosquito host[19-21], although not in all cases[16,22].

We have recently identified stable *Wolbachia* infections in natural populations of two important malaria vectors, *Anopheles gambiae* and *Anopheles coluzzii*, in Burkina Faso[23], a country with a high malaria burden. Owing to several negative reports[24-26], it was previously believed that these bacteria were unable to infect *Anopheles* species, as also suggested by the difficulties to generate stably transinfected *Anopheles* lines in the laboratory (only recently surmounted[14]). Consequently, blocking of *Plasmodium* parasites was initially demonstrated using either transient injections of non-native *Wolbachia* strains into adult *Anopheles* females[15,19], or a bird model of malaria infection transmitted by *Aedes* mosquitoes[16]. Our previous study showed that *Wolbachia* were transmitted from females to progeny with an average transmission frequency of 68% (ranging from 56 to 100%) and determined that the strain identified in *Anopheles* (which we named *w*Anga) belongs to a new arthropod-specific supergroup[23]. *Wolbachia* infections have since been confirmed in a subsequent study of *A. coluzzii* mosquitoes from the same region of Burkina Faso[27].

As *Wolbachia* can profoundly perturb insect ecology, behaviour and physiology, and can significantly reduce transmission of some human pathogens, our previous findings provide an unprecedented opportunity to assess the physiological and reproductive impacts of *w*Anga infection on anophelines and to determine how these effects influence the dynamics of malaria transmission. Here we explore the complex relationships between *Anopheles* mosquitoes, *Plasmodium* parasites and *Wolbachia* endosymbionts in natural populations. We find that *w*Anga infections are persistent in *A. coluzzii* over several years, proving these bacteria are stable residents in these mosquito populations, and are also detected in another important malaria vector, *Anopheles arabiensis*. Although crosses between infected and uninfected individuals do not show evidence of CI, infected females lay eggs more rapidly than uninfected individuals. Sampling of natural populations shows that *Wolbachia*-positive females are infected with *Plasmodium* parasites at significantly lower frequencies than *Wolbachia*-negative individuals, unveiling a negative correlation between these endosymbionts and deadly malaria parasites. Modelling the effects of these endosymbionts on malaria transmission dynamics reveals that natural *Wolbachia* infections could have a significant impact on the prevalence of malaria in human populations in Sub-Saharan Africa.

## Results

**_Wolbachia_ infections are stable and localize in the oocytes**. We returned to Vallée du Kou near Bobo-Dioulasso, Burkina Faso, in 2014 to determine the prevalence of *Wolbachia* infections in *A. coluzzii* populations. We collected 602 mosquitoes and identified infections at a frequency of 46% (275/602). Furthermore, analysis of samples collected in 2013 in the town of Soumousso found *w*Anga at a frequency of 33% (16/49) in *A. arabiensis* (Supplementary Table 1), a species not effectively targeted by LLINs and indoor residual sprays as it primarily blood feeds and rests outdoors. The prevalence of *w*Anga was moderate compared with that observed in naturally infected *Aedes* and *Culex* mosquitoes[24,28], although this may be due to difficulties in detection caused by low intensity of infection in some samples[23]. When considering all data since 2011, prevalence was variable between years (19–46%) (Supplementary Table 1), suggesting physiological or ecological factors may influence *Wolbachia* dynamics over time.

With the aim to facilitate detection of these bacteria within the female, we established a stable *w*Anga-infected mosquito line in our laboratory after colonization of *A. coluzzii* populations from the Vallée du Kou villages. We dissected ovaries from females and visualized *w*Anga at a similar frequency (43%) within the ovarian follicles by fluorescent *in situ* hybridization (FISH) experiments using a DNA probe that specifically hybridizes to *Wolbachia* 16S nucleic acid sequences (Fig. 1a). In control experiments, no signal was detected in ovaries dissected from females treated with tetracycline to clear *w*Anga infection (Fig. 1b), nor when using an excess of unlabelled probe of identical sequence in competition with the labelled probe (Fig. 1c).

**_w_Anga does not induce CI but affects oviposition**. Many *Wolbachia* strains induce CI when an infected male mates with an uninfected female, producing mostly inviable progeny. To determine whether natural *Wolbachia* infections induce CI and/or other reproductive phenotypes in field *A. coluzzii*, we collected eggs from blood-fed females in houses or larvae from breeding sites and, after adult emergence, performed crosses between virgin males and females using a force-mating technique, as field mosquitoes of this species do not mate in captivity. After mating, females were blood-fed and allowed to lay eggs individually in separate oviposition containers. We measured the number of eggs developed to assess the impact of infection on fecundity, and the larval hatch rates to measure effects on fertility and therefore detect a possible occurrence of CI. We also scored

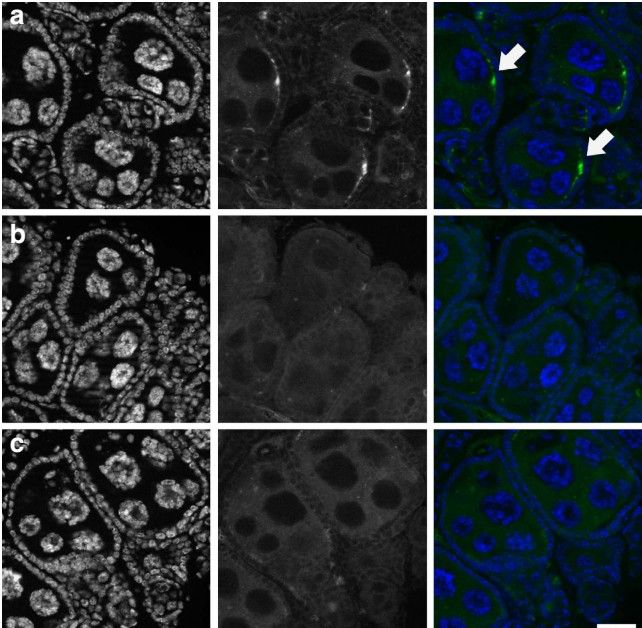

**Figure 1 | wAnga localizes to the female germline.** *w*Anga (in green, central panels) was visualized in the ovaries of 14- to 16-day-old *A. coluzzii* females by FISH. (**a**) *w*Anga is detected in ovarian follicles using a Cy3-labelled probe specific for 16S DNA (white arrows). (**b,c**) *w*Anga is absent from the follicles of tetracycline-treated control females (**b**) and in follicles of infected females in which the labelled probe was in competition with an identical unlabelled probe (1:20 labelled:unlabelled) (**c**). DNA is labelled with 4,6-diamidino-2-phenylindole (in blue, left panels). Scale bar, 20 μm.

the sex ratio of the progeny of each individual female by letting larvae grow to adulthood and counting males and females to determine whether *w*Anga induces sex distortions in these mosquitoes. Both parents of each brood were genotyped and screened for *Wolbachia* post hoc, and crosses between *Wolbachia*-positive males and *Wolbachia*-negative females (that is, those potentially inducing CI, shown in blue in Fig. 2) were compared with all other cross combinations (*Wolbachia*-positive males and females; *Wolbachia*-negative males and females; and *Wolbachia*-negative males and *Wolbachia*-positive females). Although egg inviability was slightly higher in the crosses in which CI is predicted to occur, we did not observe any significant effect of *w*Anga on egg inviability between our crosses, which suggests that this *Wolbachia* strain does not cause detectable CI in natural *A. coluzzii* populations (Kruskal–Wallis test, $\chi^2 = 1.25$, degrees of freedom (d.f.) $= 3$, $P > 0.05$; Fig. 2a). Moreover, we found no difference in the number of eggs developed (Kruskal–Wallis, $\chi^2 = 4.09$, d.f. $= 3$, $P > 0.05$; Fig. 2b), eggs laid (Kruskal–Wallis, $\chi^2 = 0.686$, d.f. $= 3$, $P > 0.05$; Supplementary Fig. 1) or in the progeny sex ratio (Student's *t*-test on $\log_{10}$-transformed ratios, $t = 0.257$, d.f. $= 26$, $P > 0.05$; Fig. 2c).

We next assessed whether egg-laying behaviour is altered by *w*Anga infection. To this end, blood-fed females from houses in the Vallée du Kou were placed in individual cups 2 days post blood feeding, and oviposition rates were counted daily. After assessing their infection status *post hoc*, we determined that *w*Anga-infected females had laid eggs significantly more quickly than uninfected females (on average $0.73 \pm 0.11$ days, log-rank test, $\chi^2 = 32.36$, d.f. $= 1$, $P < 0.0001$; Fig. 2d). This shortened oviposition timing could increase the number of gonotrophic cycles over the course of a female's lifespan, and suggests that *w*Anga-infected females may bite more frequently than

uninfected individuals (Supplementary Fig. 2), with possible consequences for malaria transmission. However, as predicted by life history theory[29], an increase in fecundity would likely induce a compensatory decrease in longevity (not measured in our experimental design), a key component for the completion of the parasite cycle within the mosquito vector.

**wAnga interferes with natural *Plasmodium* infections.** A crucial question concerning *Wolbachia* infections in *Anopheles* mosquitoes is whether these bacteria impact *Plasmodium* transmission by the mosquito vector. As Burkina Faso is a region of high malaria transmission, we reasoned that we could directly assess the effects of natural *w*Anga infection on the prevalence of *Plasmodium* parasites in blood-fed females collected from the interior of houses in the Vallée du Kou. We dissected a total of 221 blood-fed *A. coluzzii* females (genotyped *post hoc*) 5 days post collection, a time when the blood meal is fully digested and oocyst development is underway. DNA was extracted from samples containing abdomens and thoraxes (encompassing the ovaries, the midgut and the salivary glands) dissected from individual females to unravel a possible interaction between the presence of *Wolbachia* and *Plasmodium* in these individuals. A total of 116 females were positive for *w*Anga, with an infection prevalence of 52.5% (116/221). *Plasmodium* infections were detected in 12 females, producing a prevalence of 5.4% (12/221), comparable to previously reported data for this region[30]. We found a strong bias for *Plasmodium* infection in *Wolbachia*-negative individuals; 11 *Plasmodium*-positive females were negative for *Wolbachia*, and only 1 female showed co-infection with bacteria and parasites (Fisher's exact *post hoc* test on unnormalized data, two-tailed, $P = 0.0018$; Fig. 3a and Supplementary Table 2). This highly significant reduction—over 90%—suggests that *w*Anga may interfere with *Plasmodium* development in the mosquito vector, as shown in artificial *Anopheles*–*Wolbachia* combinations[14,15,19].

We went on to model whether natural *Wolbachia* infections would affect malaria transmission dynamics when these results, obtained on a limited sampling of a single transmission area, are extrapolated on a broader scale. We used a modified Ross–Macdonald model with additional compartments for *Wolbachia*-infected mosquitoes, which differ in their susceptibility to malaria (Fig. 3a) and in the length of their gonotrophic cycle (Fig. 2d), an effect that would likely impact their biting rates (Supplementary Fig. 2). We considered the *Wolbachia*-infected populations to have an identical mortality rate to uninfected mosquitoes (Fig. 3b, red line) but also ran a model where the increased speed of the gonotrophic cycle is paralleled by a decrease in female lifespan (Fig. 3b, blue line), as predicted by life history theory. Our model suggests that in both scenarios, infection with these bacteria might decrease malaria prevalence in humans, even within the range of *Wolbachia* frequencies that we observed in mosquito populations.

## Discussion

Our findings that natural *Wolbachia* infections persist in *Anopheles* mosquito populations and are negatively correlated to *Plasmodium* parasite development bring into sharp focus the potential impact of these bacteria on malaria transmission. Although our study is limited to samples from the VK5 village and we have not yet determined the effects of *w*Anga on malaria transmission by other important vectors like *A. gambiae* and *A. arabiensis*, the observed negative correlation between *Plasmodium* and *Wolbachia* prevalence in *A. coluzzii* suggests that malaria dynamics may be affected by the stable presence of these endosymbionts. Indeed, provided our results are confirmed when additional females, species and geographical locations are tested, our mathematical model predicts that increasing

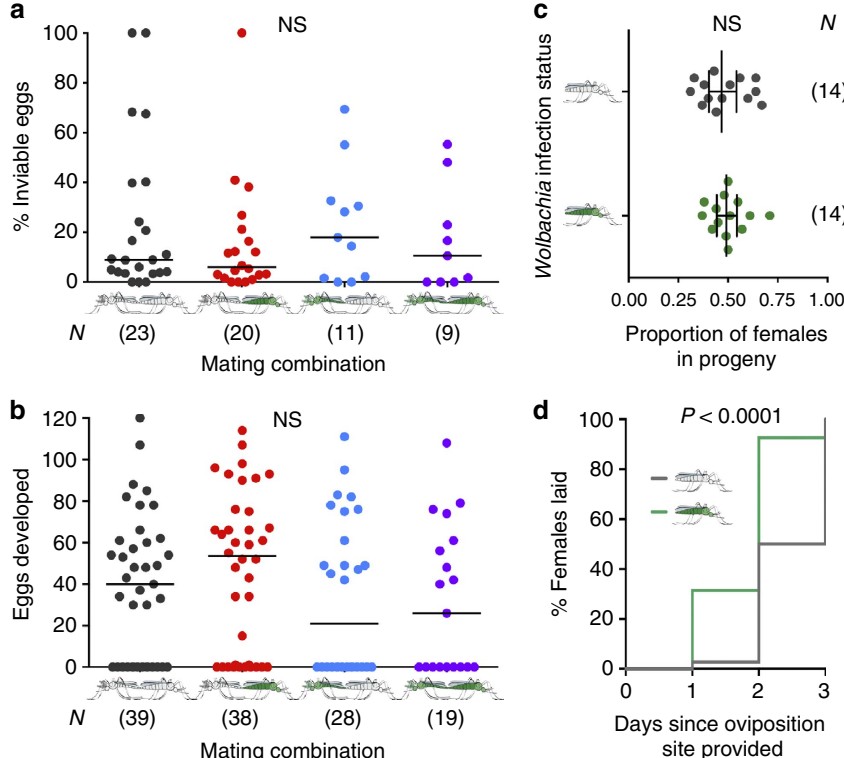

**Figure 2 | Analysis of reproductive phenotypes induced by wAnga.** (**a**–**c**) *A. coluzzii* females and males collected as eggs or larvae from Vallée du Kou were bred to adulthood and forced mated, and after blood feeding the reproductive output of females was individually scored. *w*Anga-infected (green) and -uninfected (grey) females (right) and males (left) were identified by 16S nested PCR *post hoc*, and mating couples were divided into four groups (*w*Anga-negative couples, dark grey; *w*Anga-positive couples, purple; *w*Anga-positive females mated to *w*Anga-negative males, red; and *w*Anga-negative females mated to *w*Anga-positive males, blue). (**a**) Egg inviability was calculated by dividing the number of infertile and unhatched eggs in each brood by the total number of eggs 4 days after egg laying. CI, which would manifest as higher inviability in the cross between *w*Anga-negative females and *w*Anga-positive males, was not observed (Kruskal–Wallis test, $\chi^2 = 1.25$, d.f. = 3, $P > 0.05$). (**b**) The total number of eggs produced by each female was determined by adding the number of eggs laid to the number of eggs remaining in the ovaries at the time of dissection (Kruskal–Wallis, $\chi^2 = 4.09$, d.f. = 3, $P > 0.05$). (**c**) Hatched larvae from each brood in **a** were raised to adulthood and sex was scored. The sex ratio of each brood was calculated as the number of female progeny divided by the total progeny (Student's *t*-test on $\log_{10}$-transformed ratios, $t = 0.257$, d.f. = 26, $P > 0.05$). (**d**) A total of 221 blood-fed females were collected from the walls of houses in Vallée du Kou, and 2 days after collection placed in individual oviposition containers to record the timing of egg-laying. The cumulative proportion of females that laid ($n = 143$) on each of 3 consecutive nights following access to an oviposition site is plotted. *w*Anga-infected females (green line) laid eggs more quickly than uninfected females (grey line) (log-rank test, $\chi^2 = 32.36$, d.f. = 1, $P < 0.0001$). In **a** and **b**, horizontal lines represent the medians. In **c**, vertical lines and error bars represent the geometric mean and the 95% confidence intervals, respectively. Numbers in parentheses indicate the sample size.

*Wolbachia* prevalence would reduce malaria infections in the human population, even when incorporating an increase in biting rates and regardless of a possible trade-off in lifespan. How can *w*Anga infections have such a significant impact on *Plasmodium* transmission? A number of studies have shown that strong anti-pathogenic effects likely caused by immune system activation are often associated with novel *Wolbachia* transinfections[14–16,19]. Consistent with the previously determined imperfect maternal transmission rates[23] and the lack of CI detected in this study, the observed interference with *Plasmodium* suggests that *w*Anga represents a recently introduced infection not yet fully adapted to the *Anopheles* host. Although we did not characterize localization of these bacteria in tissues besides the ovaries, we previously detected *w*Anga in *A. coluzzii* carcasses that excluded germline tissues[23]. These results suggest *w*Anga also infects somatic tissues, including those where *Plasmodium* development occurs, possibly inducing parasite killing via the upregulation of the mosquito immune system.

As hypothesized in other *Wolbachia* infections[16,31–33], *w*Anga may also effectively compete with *Plasmodium* for nutrient resources. Following blood feeding, large amounts of lipid transporters circulate to transfer lipids from the gut and the fat body to the ovary. These same lipid transporters are required by *Plasmodium* to evade the mosquito immune system[34,35]. *Wolbachia*'s potential diversion of resources may allow increased rates of immune system-mediated killing of *Plasmodium*. Alternatively, these endosymbionts may compete with other bacteria in the microbiome, especially following a blood meal[36], further draining resources away from developing *Plasmodium* parasites. Regardless of the mechanism, in future studies it will be crucial to test *Wolbachia*–*Plasmodium* interactions in different geographical locations and across different transmission seasons to determine the possible heterogeneity in pathogen-blocking effects across Sub-Saharan Africa, a factor that would have consequences for the interpretation of the outcome of key malaria control strategies such as LLINs and indoor residual sprays[3].

We found that *w*Anga infections cause limited reproductive phenotypes in the *A. coluzzii* host, and in our mating assays we did not detect CI at significant levels. Although observed in multiple *Wolbachia*-infected arthropods, CI is not a universal phenomenon, and its presence and penetrance is highly dependent on *Wolbachia* strain and host organism

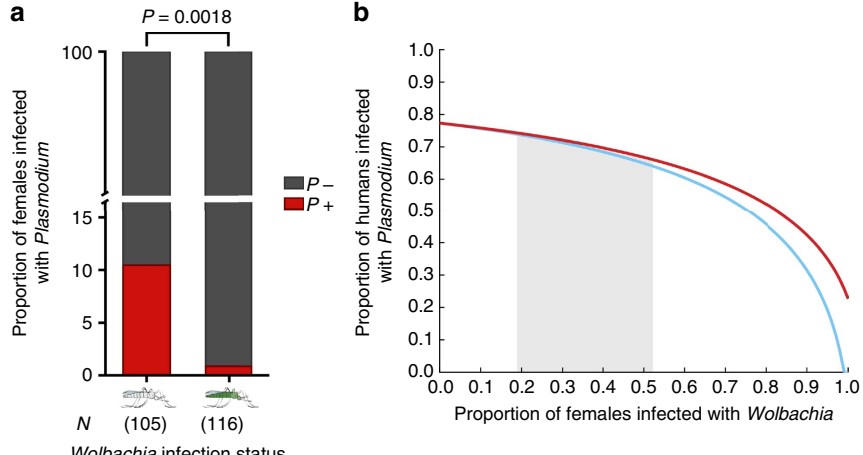

**Figure 3 | wAnga infections reduce malaria prevalence in mosquito populations and in models of human populations.** (**a**) Blood-fed females were collected from houses in Vallée du Kou, and allowed to develop eggs and oviposit. A minimum of 5 days post collection, thoraxes and abdomens were dissected and screened for the presence of wAnga and *Plasmodium* by 16S (or 16S nested) PCR and 18S quantitative PCR, respectively. Shown in red are the proportions of *Plasmodium*-infected females in both wAnga-infected (green, N = 116) and uninfected (grey, N = 105) females (Fisher's exact *post hoc* test on unnormalized data, two-tailed, P = 0.0018; Supplementary Table 2). Numbers in parentheses indicate the sample size. (**b**) A modified Ross–Macdonald model incorporating *Wolbachia*-infected mosquito compartments was used to determine the impact of *Wolbachia* on malaria prevalence. In addition to variation in biting rate and susceptibility to malaria, *Wolbachia*-infected mosquitoes were either considered to have an identical daily mortality rate to uninfected mosquitoes (red line) or an enhanced daily mortality rate due to trade-offs with increased gonotrophic cycles (blue line). The shaded area represents the range in *Wolbachia* prevalence detected in our studies across multiple years.

(reviewed in ref. 37). It remains to be established whether CI could be triggered by wAnga infection in species that do not naturally harbour these endosymbionts, as this would be an important first step towards integrating the use of wAnga into malaria control strategies. A host-dependent ability of *Wolbachia* to induce CI in non-natural hosts[37] has been demonstrated in many insects including the major dengue vector *Aedes aegypti*, in which the fruit fly wMel strain induces strong CI[38] and has successfully invaded natural populations[39]. Furthermore, *Wolbachia* densities, CI and other reproductive phenotypes can also be modulated within the same host by physiological, ecological and environmental factors including the age of the insect[40], temperature[41] and larval density[42].

Although wAnga did not increase the number of eggs laid by infected females, it increased the rate at which these eggs were laid. Over multiple gonotrophic cycles, this reproductive phenotype could increase lifetime female fecundity, although life history theory predicts that this effect would be counteracted by a trade-off with longevity[29]. The mechanism of this effect may relate to more efficient blood meal digestion, lipid deposition within the egg, egg maturation, activation of hormonal pathways that trigger oviposition[43] or a combination of these factors. Furthermore, in conditions of nutrient deprivation, wAnga could be provisioning blood-fed females with additional nutrients as shown in other *Wolbachia* infection models[44–46], ensuring timely oviposition. These hypotheses remain to be tested, and the colonized *Wolbachia*-infected *A. coluzzii* line may prove instrumental to determining the full extent of reproductive manipulations that wAnga inflicts on its mosquito host.

Combined with the previously reported identification in *A. gambiae* in the same region, the finding of wAnga infections in natural populations of *A. arabiensis*, a vector species capable of remarkable plasticity in its blood feeding and resting behaviour, suggests that this natural *Wolbachia* strain is capable of adapting to different anophelines. wAnga therefore provides a promising new tool for future malaria control strategies aimed at exophilic and exophagic *Anopheles* species not targeted by current vector control strategies.

## Methods

**Mosquito collections.** Two *Anopheles* species were collected from two separate field sites near Bobo-Dioulasso, Burkina Faso. *A. coluzzii* were collected from the VK5 village (11° 23′ N; 4° 24′ W) in the Vallée du Kou, located 30 km northwest of Bobo-Dioulasso. Mosquitoes were collected in two ways: (1) blood-fed adult females were captured in houses in the village; and (2) fourth instar larvae were collected from breeding sites in the rice fields surrounding the village and bred to adulthood in an insectary. *A. arabiensis* were collected from houses in the village of Soumousso (11° 00′ N; 4° 02′ W) located 55 km northeast of Bobo-Dioulasso as blood-fed adult females.

**Establishment of a wAnga-infected A. coluzzii colony.** wAnga-infected *A. coluzzii* mosquitoes were colonized by breeding eggs collected from VK5 to adulthood in an insectary. Adult females emerging from these eggs were crossed to *A. coluzzii* males from the Mopti colony originating from Mali (obtained through BEI Resources, NIAID, NIH: *A. gambiae*, Strain MOPTI, MRA-763, contributed by Gregory C. Lanzaro) over two generations. At each generation, after egg laying, infection in these hybrids was determined via PCR on *Wolbachia* 16S rDNA (see section 'wAnga detection by PCR and sequencing'), and the progeny of wAnga-infected mothers was pooled to establish the line. A line with the same genetic background but cleared of wAnga infection was established by treating a subset of the F3 generation with tetracycline during larval development (1 μg ml⁻¹) and adulthood (10 μg ml⁻¹).

**DNA extraction and mosquito species genotyping.** Mosquitoes were beheaded and the genomic DNA of carcasses was extracted using the Qiagen Blood and Tissue kit (Qiagen) with an extended lysis incubation step. In brief, carcasses were homogenized in 1× phosphate-buffered saline (PBS) and incubated with the kit's proteinase K and AL lysis buffer for 30 min at 56 °C. The remaining extraction steps were performed following the kit's supplementary protocol for DNA extraction of insect cells. For *A. coluzzii*, *A. gambiae* and *A. arabiensis* genotyping, the *S200* X6.1 locus from genomic DNA was amplified by PCR using standard protocols and with the following primers: forward 5′-TCGCCTTAGACCTTGCGTTA-3′, reverse 5′-CGCTTCAAGAATTCGAGATAC-3′ (ref. 47).

**wAnga detection by PCR and sequencing.** Detection of wAnga infection in mosquito carcasses was performed by PCR amplification of the 16S rDNA region using *Wolbachia*-specific primers (W-Specf: 5′-CATACCTATTCGAAGGGA TAG-3′, W-Specr: 5′-AGCTTCGAGTGAAACCAATTC-3′) and standard protocols[48]. In cases of low infection intensity, nested PCR was used for wAnga detection. A total of 2 μl of the amplification product from the initial 16S rDNA PCR was amplified using specific internal primers (16SNF: 5′-GAAGGGATAGG GTCGGTTCG-3′; 16SNR: 5′-CAATTCCCATGGCGTGACG-3′) and HotStarTaq (Qiagen) in a 20 μl reaction volume. The nested 16S rDNA PCR cycling conditions used were 15 min at 95 °C, followed by 35 cycles of 15 s at 95 °C, 25 s at 66 °C and

30 s at 72 °C, followed by 5 min at 72 °C. The sequence of the resulting 412-bp fragment was determined by Sanger sequencing. All samples sequenced were confirmed to correspond to *Wolbachia* (Supplementary Fig. 3), demonstrating that the nested PCR protocol did not generate any false positives.

**wAnga detection in mosquito oocytes by FISH.** DNA probes specific to *w*Anga 16S rDNA were designed and synthesized (5′-Cy3-CGAGGCTAAGCTAATCCC TTAAA-3′, Integrated DNA Technologies). The ovaries of *w*Anga-infected and tetracycline-treated females from the established *w*Anga colony were dissected in 1 × PBS, fixed in 4% paraformaldehyde, and treated with 80% ethanol–6% hydrogen peroxide to remove autofluorescence. FISH was then conducted using a protocol modified from Toomey *et al.*[49] Following initial pre-hybridization equilibration steps, hybridization was conducted by incubating tissues with 1 ng µl$^{-1}$ of the RNA probe in hybridization solution containing 50% formamide, 5 × SSC, 250 µg ml$^{-1}$ salmon sperm DNA, 0.5 × Denhardt's solution, 20 mM Tris-HCl and 0.1% Tween at 37 °C for 3 h. Probe competition hybridization was conducted by incubating tissues with a mixture of the labelled RNA probe (1 ng µl$^{-1}$) and an unlabelled probe of identical sequence (20 ng µl$^{-1}$). Following washes in a solution containing 1 × SSC, 0.1% Tween and 20 mM Tris-HCl then in a solution with 0.5 × SSC, 0.1% Tween and 20 mM Tris-HCl at 55 °C, tissues were blocked in 1% BSA in 1 × PBS-0.1% Tween (PBST) and stained with 1:100 mouse anti-Cy3 antibody (Santa Cruz Biotechnology). Tissues were then incubated in 1:1,000 goat anti-mouse Alexa488 antibody (ThermoFisher Scientific) and in 1 µg ml$^{-1}$ 4,6-diamidino-2-phenylindole (Sigma-Aldrich). All staining steps were followed by washes in PBST. Tissues were then mounted in Vectashield mounting media and images were acquired on a Zeiss Axio Observer inverted fluorescent microscope with Apotome2. Post-imaging processing was done using ImageJ and Adobe Photoshop CS5.

**Analysis of reproductive phenotypes.** For CI, fecundity and sex-ratio analyses, mosquitoes were collected as fourth instar larvae from breeding sites or as eggs deposited by blood-fed females collected from houses in VK5 in the Vallée du Kou. These field-collected larvae and eggs were bred to adulthood in the insectary, and adult males and females were crossed in single pairings by forced mating (protocol available on https://www.beiresources.org/Publications/MethodsinAnophelesResearch.aspx). DNA from males was immediately extracted and *w*Anga infection status was determined by 16S PCR. Females were blood-fed and placed into individual cups to allow them to oviposit individually. Following oviposition, the genomic DNA of these females was extracted and screened for *w*Anga infection by 16S PCR. Egg broods were scored for inviability (a direct indicator of CI) and the number of eggs laid (fecundity). Females were also dissected to determine the number of eggs retained and consequently the total number of eggs developed. To assess the sex ratio of the progeny, broods from these crosses were bred to adulthood and the number of males and females in each brood was determined. Infection status of males and females in each cross was determined *post hoc* by 16S PCR.

For timing of oviposition, blood-fed females were collected from houses in VK5 and put into cups 2 days post blood feeding to allow them to oviposit individually. Oviposition time was monitored by checking for the presence of eggs in each cup every day following provision of an oviposition site.

***Plasmodium* detection by quantitative PCR.** The quantitative PCR protocol published by Bass *et al.*[50] for *Plasmodium* detection was modified for use with SYBR green dye. Genomic DNA extracted from female carcasses was 10-fold diluted, and 5 µl of this dilution was quantified in a mix with 1 × Fast SYBR Green Master Mix (ThermoFisher Scientific) and 300 nM of each primer targeting a region in the *Plasmodium* 18S rDNA sequence (PlasF: 5′-CTTAGTTACGATTA ATAGGAGTAGC-3′; PlasR: 5′-GAAAATCTAAGAATTTCACCTCTGA-3′) in duplicate reactions on a StepOnePlus Real-Time PCR System (Applied Biosystems). Relative quantities were calculated using a standard curve built with serial dilutions of a plasmid containing the targeted 18S rDNA sequence. This plasmid was made by PCR amplifying the 18S rDNA region of genomic DNA extracted from the *P. falciparum* P2G12 strain with the PlasF and PlasR primers, and cloning the resulting product using the pGEM-T Vector System (Promega). Copy number quantification was determined using a Nanodrop 2000c Spectrophotometer (ThermoFisher Scientific).

**Model of feeding time.** Individual female mosquitoes were simulated stochastically through the course of their adult life. On emerging as adults, all female mosquitoes rested for 1 night before mating. The following night females took their initial blood meal, beginning gonotrophic cycles, which consisted of feeding, resting for up to 3 nights and egg laying. The number of nights spent resting before egg laying by each female in each gonotrophic cycle was determined probabilistically from the data on oviposition (Fig. 2d). This process was simulated in MATLAB R2015a (The MathWorks, Inc., Natick, MA, USA) for one million mosquitoes to determine what proportion of mosquitoes fed each night.

**Malaria transmission model.** The Ross–Macdonald compartmental model of malaria transmission[51,52] was adapted to include *Wolbachia*-infected mosquitoes,

forming a system of five delay differential equations. The infected human population ($I_H$) grew when susceptible humans received infectious bites and declined as humans recovered at a fixed rate ($r = 0.05$)[51]. Mosquito populations ($E_M$ and $E_W$) became infected after an infectious bite but were not infectious ($I_M$ and $I_W$) until the completion of the latent period, with *Wolbachia*-infected populations denoted with a W-subscript. The fraction of the mosquito population infected with *Wolbachia* ($W$) was constant throughout a single simulation.

$$\frac{dI_H(t)}{dt} = abm[I_M(t) + I_W(t)][1 - I_H(t)] - rI_H(t)$$

$$\frac{dE_M(t)}{dt} = acI_H(t)[(1 - W) - E_M(t) - I_M(t)] - \\ acI_H(t - \tau)[(1 - W) - E_M(t - \tau) - I_M(t - \tau)]e^{-u\tau} - uE_M(t)$$

$$\frac{dI_M(t)}{dt} = acI_H(t - \tau)[(1 - W) - E_M(t - \tau) - I_M(t - \tau)]e^{-u\tau} - uI_M(t)$$

$$\frac{dE_W(t)}{dt} = \hat{a}\hat{c}I_H(t)[W - E_W(t) - I_W(t)] - \\ \hat{a}\hat{c}I_H(t - \tau)[W - E_W(t - \tau) - I_W(t - \tau)]e^{-\hat{u}\tau} - \hat{u}E_W(t)$$

$$\frac{dI_W(t)}{dt} = \hat{a}\hat{c}I_H(t - \tau)[W - E_W(t - \tau) - I_W(t - \tau)]e^{-\hat{u}\tau} - \hat{u}I_W(t)$$

The *Wolbachia*-infected and -uninfected mosquitoes differed only in biting rate ($a = 0.5$ and $\hat{a} = 0.565$), susceptibility to malaria infection ($c = 0.79$ and $\hat{c} = 0.07$) and daily mortality rate ($u = 0.15$, red line[53–55]; and $\hat{u} = 0.175$, blue line in Fig. 3b). The probability of transition to humans ($b = 0.4$)[51], the length of the latent period ($\tau = 12$)[56] and the relative mosquito to human ratio ($m = 7.7$)[51] were identical between both mosquito populations. Hats denote parameters associated with *Wolbachia*-infected mosquitoes. The biting rate ($\hat{a}$) was enhanced by the relative number of blood feeds possible by *Wolbachia*-infected females compared with *Wolbachia*-uninfected females as determined in our feeding model, $\hat{a} = 1.13a$ (Supplementary Fig. 2). The susceptibility of *Wolbachia*-infected females to malaria infection ($\hat{c}$) was discounted based on the data presented in Fig. 3a, $\hat{c} = c/11$. The mortality rate ($\hat{u}$) was increased (Fig. 3b, blue line) to impose a fitness costs on *Wolbachia*-infected females, such that the average number of feeds was identical for *Wolbachia*-infected and uninfected mosquitoes. All simulations were performed in MATLAB R2015a (The MathWorks, Inc., Natick, MA, USA) using the dde23 solver.

**Data availability.** The authors declare that the data supporting the findings of this study are available within the article and its Supplementary Information files.

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

## Acknowledgements

We thank David Clarke for help with mosquito procedures, Manuela Bernardi for assistance with artwork, and Matthias Marti and Deepali Ravel for providing *Plasmodium falciparum* P2G12 genomic DNA. Research reported in this publication was supported by the National Institute Of Allergy And Infectious Diseases of the National Institutes of Health under Award Number R21AI117313 to FC. This material is based on work supported by the National Science Foundation Graduate Research Fellowship Program under Grant No. DGE1144152 to P.M. Travel to Burkina Faso was in part supported by Harvard University's 'Defeating Malaria: From the Genes to the Globe' Initiative to W.R.S.

## Author contributions

F.C., W.R.S. and P.M. designed the experiments; W.R.S., P.M. and F.B. performed the experiments; F.C., W.R.S. and P.M. analysed the data; L.M.C. and C.O.B. mathematically modelled the data; S.S., R.K.D. and A.D. provided samples for the analysis; F.C., W.R.S., P.M. and L.M.C. wrote the manuscript; W.R.S. and P.M. contributed equally to this study. The content is solely the responsibility of the authors and does not necessarily represent the official views of the National Institutes of Health.

## Additional information

**Competing financial interests:** The authors declare no competing financial interests.

