## [Peer Review File · Nature Communications]

Reviewer #1 (Remarks to the Author)

The manuscript by Shaw et al details the confirmation of Wolbachia infection in African Anopheles populations, phenotypic characterization of potential reproductive manipulations induced by Wolbachia, and some preliminary results on the potential for Wolbachia to block malaria transmission in these infected mosquitoes.

This manuscript is a significant improvement over their initial paper. However, there is some additions that they must perform before this can be published.

1. Controls used for FISH are not adequate. They need to perform competition probe, nonsense probe and anti-sense probe controls.
2. I am unconvinced by these data that Wolbachia blocks malaria infection, primarily due to the small sample size. Yes, its statistically significant, and I do think that the authors should discuss the possibility, but I think it needs to be softened. This work would be significantly strengthened if they can do experimental Plasmodium infections - they can collect mosquitoes with enough samples to conduct breeding experiments, why not for Plasmodium transmissions? If these experiments can not be conducted, I don't think it will kill the paper, but then the discussion needs to be toned down.
3. On that note, the modeling, while fine, is overkill for the low quality of the blocking data. If the authors can not conduct the blocking experiment, then I don't see the point of the model.

Reviewer #2 (Remarks to the Author)

This is an extremely interesting manuscript on a very important topic that will be of broad interest given that Wolbachia is gaining momentum as a potential new control tool for mosquito transmitted viruses and may have similar potential for malaria control. Overall the results are clear and experiments well performed. However I have a few concerns about the conclusions drawn on the impact on Plasmodium infection and possible impacts on malaria transmission as follows:

1. It is particularly surprising to me that such a strong phenotypic effect on Plasmodium is apparent when a) Wolbachia infection is described as needing nested PCR to detect and b) FISH images show low level infections in ovaries - a tissue that is normally heavily infected in most insects c) other common phenotypes such as CI are absent - which is consistent with a low level infection, as are the low maternal transmission rates.
2. The Plasmodium blocking data relies on a relatively small number of mosquitoes caught from the field and then infection status determined.

I would feel more confident in the conclusions if A) more FISH images could be displayed showing infection levels in critical somatic tissues such as gut, fat body, salivary glands that are known to be associated with pathogen blocking in other systems. B) sample sizes of field collected material could be increased to provide more confidence in association between Wolbachia and lack of Plasmodium infection or if this is not possible C) mechanistic studies using the lab colonies are undertaken showing that Wolbachia infected mosquitoes have reduced vector competence compared to tetracycline treated counterparts without Wolbachia.

Overall this has the potential to be a very significant paper if the quality of evidence could be improved. If this is not possible then I feel uncomfortable with the strength of the conclusion reached with the data presented.

Reviewer #3 (Remarks to the Author)

The manuscript reports the frequency of a naturally occurring Wolbachia in two species of Anopheline mosquitoes in Burkina Faso and through a combination of field and laboratory experiments demonstrates a negative association between Wolbachia and Plasmodium infections. It also reports a novel Wolbachia induced host effect - increased oviposition rate.

This work is incredibly significant. Artificial infection of Anopheles with Wolbachia has been a goal for decades. The recent laboratory created infection in *A. stephensi* will a demonstration of progress will not be useful for Africa. The sense among the handful of laboratories trying to create these infections was that Anophs were somehow more resistant to Wolbachia establishment than other mosquito genera. Finding Wolbachia in African vectors that are seemly very different and likely well adapted to the insect is a big win.

The measurement of a Wolbachia induced increase in the oviposition rate is also interesting. There are numerous situations where the level of CI expression cannot explain Wolbachia's persistence. It will be interesting to see if this phenotype is present in other insects or if it is unique adaptation in this genus.

Overall the manuscript is very clearly written, the statistical analysis is performed properly and the introduction of a model is compelling (although I cannot comment on its construction/effectiveness).

I have only a few suggestions.

1. Nested PCR is really sensitive to the production of false positives. Can you give any sense of whether your estimates of infection rate via FISH concur?
2. Some of your sample sizes are small - particularly from the field when you begin binning individuals into categories of Wolbachia x plasmodium infection. You should have sample sizes indicated in your figure 3 legend or on the graph. You may also mention this as a caveat in your interpretation.
3. I think if there is room at the end of the discussion it would be good for the non specialist reader to understand what the authors consider the key next steps in developing this system.

Referee #1

The manuscript by Shaw et al details the confirmation of *Wolbachia* infection in African *Anopheles* populations, phenotypic characterization of potential reproductive manipulations induced by *Wolbachia*, and some preliminary results on the potential for *Wolbachia* to block malaria transmission in these infected mosquitoes. This manuscript is a significant improvement over their initial paper. However, there is some additions that they must perform before this can be published.

We thank Referee #1 for their comments and we have addressed each of them below.

1. Controls used for FISH are not adequate. They need to perform competition probe, nonsense probe and anti-sense probe controls.

In our initial FISH experiment, we used control ovaries from tetracycline-treated females, in which *Wolbachia* levels are highly reduced/no longer detectable, and this removed the FISH signal. In response to Referee #1, we have now supplemented this control with a competition probe experiment (see Figure 1c). We feel having two negative controls is now adequate and that additional negative controls are not necessary: the competition probe provides the strongest evidence that the signal is specific to the 16S sequence from *w*Anga compared to the other methods suggested.

2. I am unconvinced by these data that *Wolbachia* blocks malaria infection, primarily due to the small sample size. Yes, its statistically significant, and I do think that the authors should discuss the possibility, but I think it needs to be softened. This work would be significantly strengthened if they can do experimental *Plasmodium* infections - they can collect mosquitoes with enough samples too conduct breeding experiments, why not for *Plasmodium* transmissions? If these experiments can not be conducted, I don't think it will kill the paper, but then the discussion needs to be toned down.

We understand that the sample size in our *Wolbachia-Plasmodium* experiments is somewhat modest and we have explicitly addressed this limitation in the revised manuscript. Unfortunately, the experiment proposed by Referee #1 to collect mosquitoes from natural populations and infect them in the laboratory is extremely challenging as females bred from natural populations do not exhibit reliable feeding behavior on membrane feeders that are needed for *Plasmodium* infection experiments. Furthermore, the availability of gametocyte donors in Burkina Faso is largely limited to peak transmission season (August-September). We have therefore followed this reviewer's suggestion and opted instead to temper our conclusions in the discussion as requested, and also in the title ("***Wolbachia* infections in natural *Anopheles* populations affect egg laying and negatively correlate with *Plasmodium* development**") and abstract.

3. On that note, the modeling, while fine, is overkill for the low quality of the blocking data. If the authors can not conduct the blocking experiment, then I don;t see the point of the model.

Although we agree that the sample size is somewhat limited and the samples are from a single geographical location, our data from over 200 females are highly significant ($p=0.0018$) and provide a *first* indication that *Wolbachia* may be playing a role in *Plasmodium* transmission in this important region. We (and referee #3) feel that the information provided by the model adds to the relevance of our findings as it shows that within the range of *Wolbachia* prevalence detected in our studies across different seasons, the observed effects of *Wolbachia* on *Plasmodium* would significantly impact malaria transmission dynamics. Once again, in the revised manuscript we have pointed out the limitations of our findings and stressed that in future studies it will be crucial to test *Wolbachia-Plasmodium* interactions in different geographical locations and possibly across different transmission seasons.

Referee #2

This is an extremely interesting manuscript on a very important topic that will be of broad interest given that *Wolbachia* is gaining momentum as a potential new control tool for mosquito transmitted viruses and may have similar potential for malaria control. Overall the results are clear and experiments well performed.

We are glad that Referee #2 found our manuscript extremely interesting, the topic very important, and the results clear.

1. It is particularly surprising to me that such a strong phenotypic effect on *Plasmodium* is apparent when a) *Wolbachia* infection is described as needing nested PCR to detect and b) FISH images show low level infections in ovaries - a tissue that is normally heavily infected in most insects c) other common phenotypes such as CI are absent - which is consistent with a low level infection, as are the low maternal transmission rates. I would feel more confident in the conclusions if A) more FISH images could be displayed showing infection levels in critical somatic tissues such as gut, fat body, salivary glands that are known to be associated with pathogen blocking in other systems.

We are grateful to this referee for this observation, as we realize that in the original manuscript we failed to provide detailed explanations for how *Plasmodium* development may be affected by *Wolbachia*. First of all, in our initial study (Baldini *et al.*, *Nature Communications* 2014) we detected *wAnga* in other tissues of *Anopheles coluzzii* mosquitoes, so it is highly likely that in the female samples collected in the current study these bacteria are residing in additional tissues besides the ovaries. We have now added this information to the manuscript. Furthermore, infection levels are not always low, as we often detect *Wolbachia* in the initial 16S PCR reaction, and we run nested PCRs to obtain sufficient DNA amounts for sequencing. We have now clarified this in the materials and methods section. A possible caveat of the FISH data is that ovaries were dissected from our *wAnga*-infected laboratory colony, where based on our observations infection levels appear less intense than in field females.

As we now discuss in the manuscript, a few hypotheses are put forward in the literature that may explain the negative correlation between *Wolbachia* and *Plasmodium*. Firstly, newly introduced infections may induce an upregulation of the immune system as demonstrated in a number of transinfection studies (Kambris *et al.*, *PLoS Pathogens* 2010; Bian *et al.*, *Science* 2013; Hughes *et al.*, *PLoS Pathogens* 2011, Moreira *et al.*, *Cell* 2009).

A second hypothesis is that *wAnga* may effectively compete with *Plasmodium* for nutrient resources in the mosquito. Following blood-feeding, large amounts of lipid transporters (Lipophorin, Vitellogenin) circulate to transfer lipids to the ovary. These same lipid transporters are required by *Plasmodium* to evade the mosquito immune system (Mendes *et al.* *PLoS Pathogens* 2008, Rono *et al.*, *PLoS Biology* 2010). *Wolbachia*'s potential diversion of resources would allow increased rates of immune system-mediated killing of *Plasmodium*. *Wolbachia* infection can also decrease cholesterol in *Aedes* mosquitoes (Caragata *et al.*, *Microbial Ecology* 2014), which is required for membrane production during *Plasmodium* and *Wolbachia* cellular division, and has been shown to modulate the pathogen-blocking ability of the bacteria (Caragata *et al.*, *PLoS Pathogens* 2013). Additionally, competition between *Wolbachia* and other bacteria of the microbiome, especially following a blood meal (Hughes *et al.* *PNAS* 2014), may further drain resources away from developing *Plasmodium* parasites.

Pathogen-blocking is very much dependent on the *Wolbachia* strain and host species interaction. For example, *wAlbB* blocks *P. falciparum* (Hughes *et al.* *PLoS Pathogens* 2011) but enhances *P. berghei* infection (Hughes *et al.* *Applied and Environmental Microbiology* 2012), making it impossible to generalize findings from previous interactions to *wAnga* infections of *Anopheles*. Although less likely, *wAnga*, a highly divergent strain of dipteran *Wolbachia* (Baldini *et al.* *Nature Communications* 2014), may encode unknown and highly efficient mechanisms of pathogen-blocking that are not found in other strains, and may be able to block pathogen development even at relatively low infection intensities.

It is important to note that field-caught females support relatively low *Plasmodium* oocyst numbers within the midgut (<5) (reviewed in Tripet *et al. Trends in Parasitology* 2008) and do not resemble laboratory infections of *Plasmodium falciparum*, where oocysts can number in the hundreds. This implies that the success threshold for *Plasmodium* infection is higher in natural conditions than in the laboratory and, therefore, small changes in immune gene expression or resource diversion caused even by low-level *w*Anga infections could have a drastic negative effect on the parasite.

We now present most of these hypotheses in our discussion.

2. The *Plasmodium* blocking data relies on a relatively small number of mosquitoes caught from the field and then infection status determined. I would feel more confident in the conclusions if B) sample sizes of field collected material could be increased to provide more confidence in association between *Wolbachia* and lack of *Plasmodium* infection or if this is not possible C) mechanistic studies using the lab colonies are undertaken showing that *Wolbachia* infected mosquitoes have reduced vector competence compared to tetracycline treated counterparts without *Wolbachia*.

Despite the limitations of our sample size, we remain confident that our experimental set-up based on the analysis of *Plasmodium* infections in females collected from houses remains the most appropriate and relevant way to determine the effects of natural *Wolbachia* infections on malaria transmission. We believe laboratory infections carried out with a single *Plasmodium* isolate would not strengthen our data as they would not represent the complexity of field infections (as recently shown for instance by Elderling *et al., Scientific Reports* 2016). *Plasmodium* infection intensity in laboratory infections is also much higher than that observed in natural populations and may overwhelm what may be a subtle pathogen blocking phenotype in a highly tuned system. Given additional field samples would only become available during the August-September malaria transmission season, further analyses would have to be delayed until that time. We have, therefore, stressed the limitations of our studies throughout the revised manuscript, in the abstract, results and discussion. This is also reflected in our changes to the title, which now presents a more cautious interpretation of our data and reads “***Wolbachia* infections in natural *Anopheles* populations affect egg laying and negatively correlate with *Plasmodium* development.**”

Overall this has the potential to be a very significant paper if the quality of evidence could be improved. If this is not possible then I feel uncomfortable with the strength of the conclusion reached with the data presented.

In an effort to address these concerns, we have toned down the strength of our conclusions and have indicated potential limitations with the data.

Referee #3

This work is incredibly significant.... The measurement of a *Wolbachia* induced increase in the oviposition rate is also interesting. There are numerous situations where the level of CI expression cannot explain *Wolbachia*'s persistence. It will be interesting to see if this phenotype is present in other insects or if it is unique adaptation in this genus. Overall the manuscript is very clearly written, the statistical analysis is performed properly and the introduction of a model is compelling (although I cannot comment on its construction/effectiveness). I have only a few suggestions.

We are glad that the reviewer thinks our manuscript is interesting and incredibly significant. We believe we have addressed all the Referee's comments in this revised manuscript.

1. Nested PCR is really sensitive to the production of false positives. Can you give any sense of whether your estimates of infection rate via FISH concur?

In order to rule out possible false positives identified in nested PCR, we have now sequenced the amplification products of 116 reactions and confirmed the identity of *Wolbachia* DNA in 100% of samples. We have included a supplementary figure of an alignment of 20 of these sequenced *wAnga* 16S rDNA products with *wAlbB* and *wPip* reference sequences (Supplementary Figure 3) and have added this information to the materials and methods. We also provide estimates of infection rates via FISH (43%), and confirmed that those are similar to the ones reported by PCR.

2. Some of your sample sizes are small - particularly from the field when you begin binning individuals into categories of *Wolbachia* x plasmodium infection. You should have sample sizes indicated in your figure 3 legend or on the graph. You may also mention this as a caveat in your interpretation.

We agree with Referee #3. We initially provided the sample sizes to Figure 3 as Supplementary Table 2 in our original submission but have now also added this information to Figure 3 and its legend. We have also discussed the limitation of our sample sizes in the text, and tempered the strength of our conclusions accordingly.

3. I think if there is room at the end of the discussion it would be good for the non specialist reader to understand what the authors consider the key next steps in developing this system.

We agree with this suggestion and have expanded our discussion to include the implications of our findings for the use of *Wolbachia* for malaria control, specifically outlining the possibility of introducing *wAnga* into other *Anopheles* species.

Reviewer #2 (Remarks to the Author)

I feel that the modified manuscript adequately addresses earlier criticisms and makes an important and exciting contribution to the emerging field of Wolbachia interactions with anopheline mosquitoes and Plasmodium. The authors should be congratulated on their work.

Reviewer #3 (Remarks to the Author)

I am happy that the authors have done everything possible within the framework of their sampling system to address the concerns of the reviewers. I think the work is now suitable for publication.

Reviewer #2

I feel that the modified manuscript adequately addresses earlier criticisms and makes an important and exciting contribution to the emerging field of Wolbachia interactions with anopheline mosquitoes and Plasmodium. The authors should be congratulated on their work.

We thank Reviewer #2 for their comments and their praise of the work. We are pleased they feel our modified manuscript adequately addresses earlier criticisms.

Reviewer #3

I am happy that the authors have done everything possible within the framework of their sampling system to address the concerns of the reviewers. I think the work is now suitable for publication.

We thank Reviewer #3 for their comments and are pleased they feel we have done our best to address their earlier concerns. We are grateful that they now feel the work is suitable for publication.